# A Novel Association between YKL-40, a Marker of Structural Lung Disease, and Short Telomere Length in 10-Year-Old Children with Bronchopulmonary Dysplasia

**DOI:** 10.3390/children8020080

**Published:** 2021-01-24

**Authors:** Ewa Henckel, Anna James, Jon R Konradsen, Björn Nordlund, Malin Kjellberg, Eva Berggren-Broström, Gunilla Hedlin, Sofie Degerman, Kajsa Bohlin

**Affiliations:** 1Department of Clinical Science, Intervention and Technology, Karolinska Institutet, 141 86 Stockholm, Sweden; kajsa.bohlin@ki.se; 2Department of Women’s and Children’s Health, Karolinska Institutet, 171 77 Stockholm, Sweden; jon.konradsen@sll.se (J.R.K.); bjorn.nordlund@ki.se (B.N.); malin.kjellberg@sll.se (M.K.); Gunilla.Hedlin@ki.se (G.H.); 3Department of Neonatology, Astrid Lindgren Children’s Hospital, Karolinska University Hospital, 141 86 Stockholm, Sweden; 4Department of Experimental Asthma and Allergy Research, Institute of Environmental Medicine, Karolinska Institutet, 171 77 Stockholm, Sweden; anna.james@ki.se; 5Lung and Allergy Unit, Astrid Lindgren Children’s Hospital, Karolinska University Hospital, 171 76 Stockholm, Sweden; 6Department of Clinical Science and Education, Södersjukhuset, Karolinska Institutet, 118 83 Stockholm, Sweden; eva.berggren-brostrom@sll.se; 7Department of Pediatrics, Sachs’ Children’s and Youth Hospital, 118 83 Stockholm, Sweden; 8Department of Clinical Microbiology, Umeå University, 907 36 Umeå, Sweden; sofie.degerman@medbio.umu.se; 9Department of Medical Biosciences, Umeå University, 907 36 Umeå, Sweden

**Keywords:** telomere length, YKL-40, bronchopulmonary dysplasia, preterm, biomarker, SPECT, lung function, inflammation-accelerated aging, oxidative stress, V/Q ratio

## Abstract

Extremely preterm infants are born with immature lungs and are exposed to an inflammatory environment as a result of oxidative stress. This may lead to airway remodeling, cellular aging and the development of bronchopulmonary dysplasia (BPD). Reliable markers that predict the long-term consequences of BPD in infancy are still lacking. We analyzed two biomarkers of cellular aging and lung function, telomere length and YKL-40, respectively, at 10 years of age in children born preterm with a history of BPD (*n* = 29). For comparison, these markers were also evaluated in sex-and-age-matched children born at term with childhood asthma (*n* = 28). Relative telomere length (RTL) was measured in whole blood with qPCR and serum YKL-40 with ELISA, and both were studied in relation to gas exchange and the regional ventilation/perfusion ratio using three-dimensional V/Q-scintigraphy (single photon emission computer tomography, SPECT) in children with BPD. Higher levels of YKL-40 were associated with shorter leukocyte RTL (Pearson’s correlation: −0.55, *p* = 0.002), but were not associated with a lower degree of matching between ventilation and perfusion within the lung. Serum YKL-40 levels were significantly higher in children with BPD compared to children with asthma (17.7 vs. 13.2 ng/mL, *p* < 0.01). High levels of YKL-40 and short RTLs were associated to the need for ventilatory support more than 1 month in the neonatal period (*p* < 0.01). The link between enhanced telomere shortening in childhood and structural remodeling of the lung, as observed in children with former BPD but not in children with asthma at the age of 10 years, suggests altered lung development related to prematurity and early life inflammatory exposure. In conclusion, relative telomere length and YKL-40 may serve as biomarkers of altered lung development as a result of early-life inflammation in children with a history of prematurity.

## 1. Introduction

Bronchopulmonary dysplasia (BPD) is the most common severe complication affecting children born extremely preterm. It is a chronic lung disease associated with early mechanical ventilation, supplemental oxygen, and inflammation that may lead to impaired lung growth and long-term effects, such as reduced lung capacity, pulmonary hypertension, emphysema, and neuro-cognitive problems [1,2,3,4]. We still lack good biomarkers to predict the future morbidity of BPD patients after infancy.

The cause of BPD is multifactorial and the development of the disease is driven by inflammation [3]. Extremely preterm children are born with immature lungs in a storm of cytokines [5], which in combination with long periods of supplementary oxygen treatment exposes them to oxidative stress. There is experimental evidence that oxygen disrupts the development of human fetal lung mesenchymal cells, which can contribute to lung growth arrest, a characteristic feature of BPD [6].

Telomeres are non-coding DNA sequences that stabilize chromosomal ends, and telomere length is a marker of cellular aging [7]. Oxidative stress and inflammation, as well as lung aging in obstructive pulmonary disease, are associated with short telomere length [8,9,10]. Pediatric studies of telomere length as a biomarker of disease are rare, particularly in relation to prematurity, but in a large study of preterm-born adolescents, Hadchouel et al. showed that shorter salivary telomere length was associated with impaired lung function [11].

The chitinase-like protein YKL-40, also called chitinase-3-like protein 1 (CHI3L1), is a potential serum biomarker associated with airway remodeling and vascular smooth muscle proliferation and migration [12]. Elevated levels of YKL-40 are positively correlated with the severity of asthma, implying impairment in lung function, which has been described in adults [13,14,15], as well as in the pediatric population [12,16,17].

Markers of lung tissue damage in relation to prematurity are scarce. Our group recently showed elevated levels of YKL-40 in serum when comparing school-aged children with bronchopulmonary dysplasia to children with allergic asthma, and this could be a possible marker of remodeling and inflammation following BPD [18]. In the same cohort, we found that the relative telomere length (RTL) was similar in the preterm-born children with BPD and term-born children with asthma, but a shorter RTL was associated with impaired lung function and male sex, irrespective of prematurity [19].

The long-term consequences of BPD may include impaired lung function, particularly involving the small airways, but this is difficult to evaluate with standard methods such as dynamic and static spirometry. Single photon emission computer tomography (SPECT) is a technique used to quantify functional loss by measuring how ventilation and perfusion in different regions of the lung match. It has previously been shown that lung volume with good V/Q matching is lower in children with BPD [20,21] than in healthy individuals [22].

We hypothesized that children born with immature lungs exposed to an environment leading to inflammation in the neonatal period will develop structural tissue damage and signs of cellular aging later in life. Therefore, we evaluated YKL-40 and relative telomere length (RTL) at ten years of age in a cohort of extremely preterm-born children with BPD.

## 2. Materials and Methods

### 2.1. Materials

#### Study Cohort

Patients included in this study were part of a larger follow-up study of preterm children with BPD, the Premature follow-up with Lung function Mannitol and Methacholine (PULMM) study. Detailed information about the clinical characteristics, lung function (dynamic and static spirometry), inflammation (cytokines and fractional exhaled nitric oxide, FeNO), and neonatal lung morbidity has previously been reported [23] and described in relation to the relative telomere length [19] and YKL-40 [18]. The ventilation to perfusion ratio (V/Q ratio) using SPECT has been reported for twenty-six children with BPD from the original PULMM study [21].

In the present cross-sectional study, a cohort of twenty-nine preterm-born 10-year-old children from the original PULMM cohort was included based on the availability of blood samples, Figure 1. The diagnosis of BPD was defined as supplementary oxygen requirement for more than 28 days of life. Disease severity grading was performed at a postmenstrual age of 36 weeks: mild BPD if no supplemental oxygen was required; moderate if less than 30% oxygen was required; and severe if more than 30% oxygen and/or ventilatory support was required [24].

No sample size calculations were performed, since this was a predefined study cohort. Parental written informed consent was obtained. The Regional Ethical Review Board in Stockholm approved the PULMM study (2008/1677-31/4, 2009/684-32, 2009-06-04) with an amendment for telomere analyses (2012/679-32, 2012-05-07) and for SPECT study examination (2009/176-32, 2016/913-32, 05/2009, K2743-2016).

### 2.2. Telomere Length

DNA was extracted from whole blood, and relative telomere length (RTL) was determined by quantitative PCR, as previously described [19,25]. In short, the T/S (telomere (TEL)/single copy gene (HBB)) values were calculated by the 2^−ΔCt^ method, where ΔCt = Ct_TEL_ − Ct_HBB_. Relative telomere length (RTL) values were obtained by dividing the sample T/S values with the T/S value of the reference sample (CCRF-CEM DNA), which was included in each run.

#### YKL-40

The serum level of YKL-40 was analyzed by ELISA, as previously described [16,18,26]. Briefly, levels were measured by ELISA in accordance with the manufacturers’ instructions (Human Chitinase 3-like 1 DuoSet ELISA Development Kit, RnD Systems, Abdingdon, UK). Two different dilutions were made for each sample from which an average was obtained, and all samples were analyzed in duplicate, in a random order.

### 2.3. SPECT Examination

Twenty-five children out of the 29 children in the BPD group were examined with three-dimensional single photon emission computed tomography ventilation perfusion scintigraphy (TRIAD XLT, Trionix Research Laboratory, Twinsberg, OH, United States) using a protocol described in previous studies [20,27]. Perfusion was mapped by intravenously injected Technetium 99 m labelled macro aggregate albumin (Tc^99m^-MAA; Mallinckrodt Medical, Petten, The Netherlands) followed by inhalation of Technegas aerosol (Tetley Manufacturing Ltd. Sydney, Australia) to record ventilation during normal tidal breathing, keeping the child lying in the same position. Ventilation perfusion ratios (V/Q ratios) were calculated within the lung at regional levels. A V/Q ratio between 0.6 and 1.4 was defined as matched. Areas with V/Q > 1.4 were defined as mismatched and regions with V/Q < 0.6 as reverse mismatched. In mismatched areas, perfusion defects contributed to the abnormalities to a greater extent as opposed to regions with reverse mismatch, where reduced ventilation was more prominent [21].

### 2.4. Statistical Analysis

The relationship between YKL-40 and RTL was estimated using an ordinary least-squares linear regression. As the sample size was small (*n* = 29), the residuals were analyzed to identify deviations from normality with no relevant irregularities found. Pearson correlation coefficients were used for the correlation analysis. Correlations adjusted for sex, age at the time of blood sampling and for gestational age at birth were estimated by regressing the YKL-40 and RTL on age and sex and estimating the correlations among the residuals. An attenuation analysis was done to investigate the degree to which neonatal lung morbidity explained the relationship between YKL-40 and RTL. The morbidity variables (duration of supplemental oxygen, days on continuous positive airway pressure (CPAP), days on ventilator, surfactant administration, patent ductus arteriosus, retinopathy of prematurity and severity of bronchopulmonary dysplasia (BPD)) were added to the model both individually and cumulatively. Data with a normal distribution are reported as means and standard deviations, and data with non-normal distribution are reported as medians and interquartile ranges. The statistical significance was set to *p* < 0.05. Analyses were performed using STATA Version 15 (StataCorp LLC, College Station, TX, USA) and IBM SPSS Statistics version 27 (IBM Corporation, Armonk, NY, USA).

## 3. Results

### 3.1. Clinical Characteristics

Detailed characterization of the BPD and the asthma group have been reported previously [19] and are summarized in the Appendix A.

### 3.2. Markers of Lung Disease

There was a statistically significant correlation between shorter RTLs and higher levels of serum YKL-40 (*r* = −0.55, *p* = 0.002) in the BPD group (children born preterm), as shown in Figure 2.

The correlation remained robust after adjusting for gestational age at birth, sex, age at follow-up and SPECT (Table 1) and no associations were found between lung function parameters from dynamic and static spirometry at ten years of age (Appendix A).

The association between RTL and YKL-40 was not affected by neonatal lung morbidity (days on supplementary oxygen, days on CPAP or days with mechanical ventilation). However, more severe respiratory morbidity in the neonatal period, here defined as duration of ventilatory support for more than one month (mechanical ventilation + CPAP more than 28 days), was associated with higher YKL-40 (19.3 ± 6.1 versus 11.9 ± 3.2, *p* < 0.01) and shorter RTL (1.57 [0.3] versus 1.65 [0.4], *p* > 0.05), shown in Figure 3.

Normal values of RTL and YKL-40 are lacking. The correlation we found between high values of YKL-40 and short RTLs in the BPD group was not detected in the asthma group (term-born children), as shown in the Appendix A.

### 3.3. SPECT Ventilation/Perfusion

No associations were found between the percentage of lung volume with matched V/Q ratio and levels of YKL-40 or short RTLs. We could not find any difference when comparing the group with good ventilation perfusion matching (>70% of the total lung volume) and those individuals with a higher degree of abnormalities in ventilation and perfusion distribution and matching (matched V/Q ratio of less than 70% within the lung). The V/Q mismatch was not associated with high values of YKL-40 or short RTLs, as shown in Figure 2.

## 4. Discussion

To our knowledge, this is the first time that an association between cellular aging and signs of structural lung disease has been described in children. Children born preterm with a history of BPD were found to have impaired lung function with higher levels of YKL-40 related to neonatal respiratory morbidity and inflammation, which was indicative of structural lung disease, at ten years of age. In that same cohort, the relative telomere length was similar to that of their term-born peers, but short telomeres were associated with impaired lung function irrespective of prematurity [18,19,23]. In this study, we found that a higher level of YKL-40 correlated with a shorter relative telomere length in children growing up with BPD and was associated with longer periods of ventilatory support in the neonatal period. These signs of airway remodeling associated with cellular aging at ten years of age indicate that inflammation and oxidative stress may play a part in the long-term consequences of extreme prematurity. However, we could not establish any association between YKL-40 and RTL with ventilation and perfusion mismatch as a sign of alveolar-capillary impairment in this extremely preterm-born group of ten-year-old children using V/Q-scintigraphy (SPECT).

The long-term reduced lung capacity consequences of children born preterm have been well studied, but we still do not understand the exact mechanism by which this lung morbidity occurs [28]. Nordlund et al. described the hallmarks of BPD as reduced lung function, impaired diffusion capacity, and hyper-reactive airways in children born extremely preterm compared with children born at term who developed allergic asthma during childhood [23]. Exhaled nitric oxide (FeNO) was found to be significantly lower in children with BPD, and the impaired lung function could not be explained by ongoing inflammation. Baraldi et al. found low exhaled nitric oxide concentrations in school-age children with BPD, indicating less inflammation measured as FeNO in spite of a reduced lung capacity [29], supporting our findings in this cohort. Neonatal respiratory morbidity did not affect the correlation between YKL-40 and RTL, but, as previously reported [18], YKL-40 was present in a significantly higher concentration when treatment with mechanical ventilation and CPAP exceeded one month. Impaired lung function with affected indices of dynamic spirometry was evident in the preterm group (Appendix A), and only 62% of the children with SPECT examination had good V/Q matching [21]. However, no correlation was found between V/Q ratio and our analyzed biomarkers of structural lung damage and cellular aging at ten years of age.

Lungs develop postnatally, and markers of lung tissue remodeling are needed to individualize treatment and optimize the development of immature lungs. A short telomere length is a marker of obstructive lung disease in adults [30], but this has not yet been shown convincingly in young children and adults born preterm [11]. Telomere length attrition varies over time with a higher attrition rate early in life, but longitudinal studies in relation to disease are scarce [31]. Our group evaluated the relationships of cellular aging with relative telomere length and DNA methylation in a small longitudinal study of relatively healthy and mature preterm-born infants compared with term-born controls and found no accelerated aging the two first years of life [25]. Preterm-born infants have longer telomeres than term-born infants [32], and telomere attrition is faster in the first four years of life [33]. In this study, we only had one time point at which we could evaluate RTL in two groups of children with lung disease: BPD versus a group with asthma. The lack of a healthy control group might have hidden any differences in telomere length due to inflammation and oxidative stress exposure during the neonatal period, and, to our knowledge, the impact of asthma development during childhood on telomere shortening has not yet been studied.

YKL-40 is a marker of airway remodeling [34] and was studied in relation to hyperoxia-induced acute lung injury in mice by Sohn et al. They found that hyperoxia inhibits YKL-40, which serves as a regulator of inflammation and an important inhibitor of oxidant-induced lung injury permeability and epithelial apoptosis in the murine lung. Interestingly, they also measured YKL-40 in tracheal aspirates from nine preterm-born children (mean gestational age week 26) on the second day of life. Five children who developed severe BPD or died had significantly lower levels of YKL-40 compared to four children who also needed mechanical ventilation as treatment for respiratory distress but did not need oxygen at a postmenstrual age of 36 weeks [35]. Lee et al. speculated that physiological levels of YKL-40 protect tissue from oxidative damage, but low levels may increase hyperoxic injury, and when levels of YKL-40 are too high, structural changes and inflammation occur [34]. We still lack reference values for normal concentrations of serum YKL-40.

YKL-40 is not a good predictor for future development of asthma in pre-school children with wheeze [26,36], and, in a prospective study of 170 term-born infants, no significant association was found between YKL-40 levels in cord blood and lung function and asthma development at 6 years of age. In a cohort of 83 extremely preterm infants with BPD, the serum YKL-40 concentration could not predict pulmonary hypertension [37]. On the other hand, YKL-40 is a known marker of severe therapy-resistant asthma in children [16], and, in a longitudinal study of acute wheeze in children aged between 6 and 44 months, higher levels of YKL-40 were found to have a positive correlation with blood neutrophil counts [26]. James et al. found significantly higher levels of YKL-40 in the BPD group compared to the group with well-controlled asthma in spite of there being no signs of acute ongoing inflammation [18].

This study has several limitations due to its cross-sectional design and because it was not powered or designed to examine the relationship between the structural tissue damage protein YKL-40 and RTL. The lack of a healthy control group and the small sample size meant that we had to interpret any results with caution. Despite these limitations, we show a novel association between blood telomere length and serum YKL-40 in relation to neonatal lung disease and impaired lung function in a well-characterized cohort of extremely preterm-born children. Understanding the consequences of early life inflammation contributes to the tailoring of treatment with the aim of enhancing organ development postnatally without tissue damage and thereby avoiding the long-term consequences of chronic lung disease. Further studies are needed to elucidate whether YKL-40 and telomere length can be used as biomarkers of altered lung development and be used to predict the long-term consequences of BPD.

## 5. Conclusions

The link between enhanced telomere shortening in childhood and structural remodeling of the lungs seen in children with former BPD suggests altered lung development related to prematurity and early life inflammatory exposure.

## Figures and Tables

**Figure 1 children-08-00080-f001:**
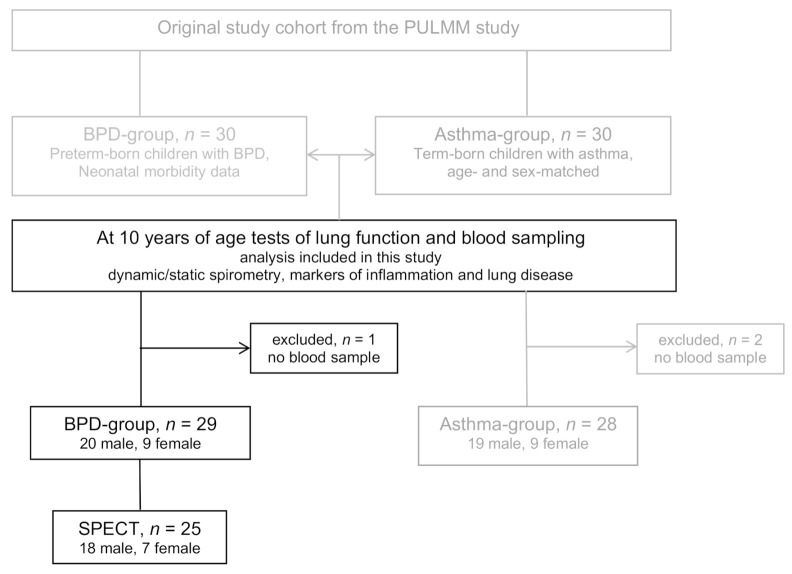
Flow chart of eligible patients from the original cohort of the PULMM study (PrematUre follow up with Lung function Mannitol and Methacholine). The final cohort included 29 children born preterm with bronchopulmonary dysplasia (BPD) at ten years of age. Lung function with spirometry, YKL-40 and relative telomere length (RTL) was analyzed in all children, and three-dimensional V/Q-scintigraphy (single photon emission computer tomography, SPECT) was analyzed in a subgroup of 25 children.

**Figure 2 children-08-00080-f002:**
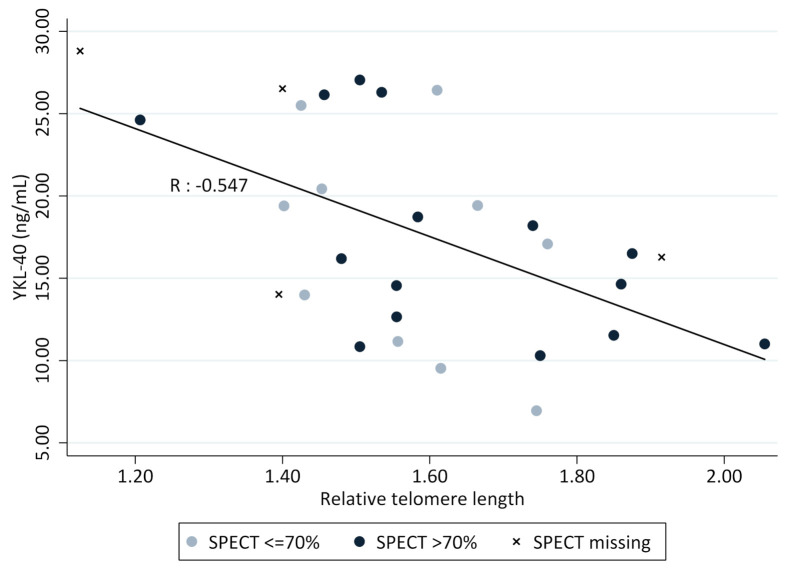
Correlation between relative telomere length (RTL) and YKL-40 at 10 years of age in 29 children born preterm with a history of BPD (*p* < 0.01). The darker circles represent good ventilation to perfusion ratios in more than 70% of the total lung volume (*n* = 15), and the brighter circles (*n* = 10) represent the group with more abnormalities (<70%).

**Figure 3 children-08-00080-f003:**
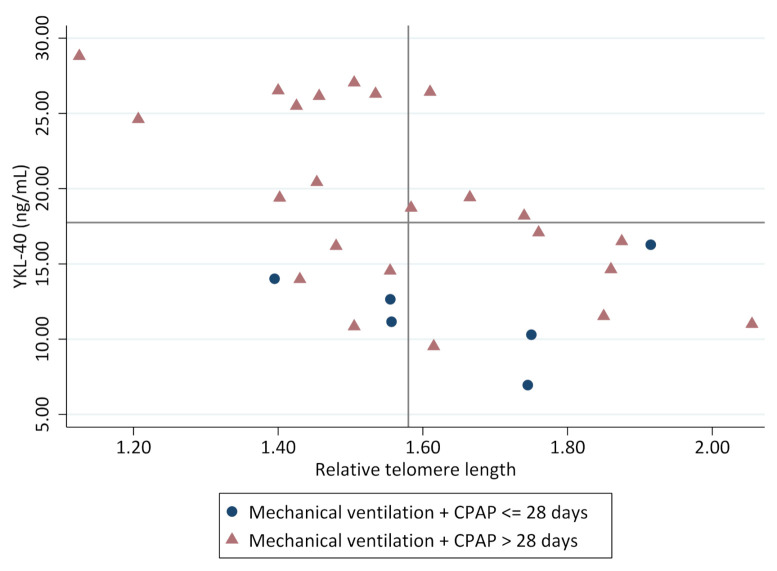
Markers of structural lung damage and cellular aging at ten years of age in children born preterm with BPD in relation to the need for ventilatory support of more than one month in the neonatal period. The lines in the graph represent the median for RTL (1.58) and for YKL-40 (17.8 ng/mL).

**Table 1 children-08-00080-t001:** Correlation between the relative telomere length (RTL) and YKL-40 at ten years of age in children born preterm with a history of BPD.

Ordinary Least Squares Regression, *n* = 29	
	Coefficient (95% CI)
RTL-YKL-40	−0.02 [−0.03; −0.007] **
adjusted for:	
sex	−0.02 [−0.03; −0.005] **
age at testing	−0.02 [−0.03; −0.007] **
SPECT	−0.02 [−0.03; −0.002] *
gestational age at birth	−0.02 [−0.03; −0.007] **

Median (interquartile range); * significance *p* < 0.05; ** significance *p* < 0.01; RTL = relative telomere length (ratio of ΔCt values (ΔCt = Ct_sample_ − Ct_referenceDNA_)); YKL-40 = chitinase-3-like protein1 (ng/mL). SPECT = single photon emission computer tomography.

## Data Availability

The data presented in this study are available on request from the corresponding author. The data are not publicly available due to ethical reasons.

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
