# Peer review of "A Novel Association between YKL-40, a Marker of Structural Lung Disease, and Short Telomere Length in 10-Year-Old Children with Bronchopulmonary Dysplasia"

_children, 2021, doi:10.3390/children8020080_

Round 1
Reviewer 1 Report
The paper shows interesting results about the elevation of YKL-40 level and telomere shortening.
Inclusion of a control group rather than asthma group will give a better idea about the correlation with SPECT data.
Author Response
Thank you for a very valid comment. We are aware of the weakness of not having a healthy control group. The present study was performed in a pre-excisting cohort of 29 preterm born children with BPD at ten years of age, in which the original study protocol included lung function testing and provocation tests with methacholine and mannitol. As it was not found ethical to expose healthy children for that provocation, a control group of term born children with asthma was included.
Similarly regarding the SPECT data, we did not have ethical permission to expose healthy children for unnecessary radiation involved in the examination, even though the radiation dose in SPECT is low.
We have revised the manuscript focusing on the comparisons within the the BPD group and therefore hope that the obvious limitations with the control group will be less concerning.
Reviewer 2 Report
Henckel et al provide evidence that increased expression of the chitin-like-protein YKL-40 correlates with shortened Telomere length and V/Q mismatch, an indicator for structural changes in the lung, in 10-year old children.
Concerns: data presentation
All tables: it is easier to read the left column using left alignment for text
Table 2: The numbers for markers of lung tissue damage are ratios of ΔCt values; this should be added to the legend, not only the methods.
Figure 1: add the correlation values to the graph. Seeing the figure and the correlation index makes it clearer. I don’t see the need for figure 1b, black dots are the same in both a and b. Or keep the asthma and BPD group separate.
Figure 2: not clear to me. It might be better to separate the information into two figures with the higher and lower V/Q ratios. The association between lung function and YKL-40 and RTL as indicated in the abstract should be visible in addition to the YKL-40 RTL correlation.
Author Response
All Tables: Response 1
We agree and have adjusted all tables as suggested.
Table 2: Response 2
Table 2 has been revised to clarify that the association between RTL and YKL-40 stayed robust when adjusting for gestational age at birth, age at testing, sex and for SPECT data. As kindly suggested by the reviewer, we have also clarified abbreviations and added the unit of YKL-40 (ng/mL) and for RTL (the ratios of values Ct (Ct = Ctsample – CtreferenceDNA)).
Figure 1: Response 3
We agree with the reviewer that there was redundant information in Figure 1. To clarify, the correlation values have been incorporated into the graph and the figure revised.
Figure 1A has been omitted and moved to the supplementary material as Figure S1.
Figure 1 now represent only the BPD group. Information of SPECT values < 70% of normal and > 70% of normal matched ventilation to perfusion ratio have been added in the graph in order to visualize that higher and lower V/Q ratios was not associated to the RTL/YKL-40 correlation.
Figure 2: Response 4
We are very grateful for the constructive comment to help us clarify the scope of the manuscript and agree that Figure 2 did not correctly convey our message.
The correlation between RTL and YKL-40 was not associated to neither lung function at ten years of age, nor to alveolar-capillary impairment measured as V/Q ratios. The detailed information regarding V/Q matching and lung function values have therefore been summarized in the supplementary Table S1.
However, there was an association of these markers of lung tissue remodeling (YKL40) and cellular ageing (RTL) to more severe neonatal respiratory morbidity defined as the need for ventilatory support > 1 month in the neonatal period. This association was not explained by gestational age at birth, nor by sex or age at testing.
We have done major revisions in the text to more clearly describe our findings and also developed a new Figure 2. The data of V/Q ratios have been replaced and are now incorporated in Figure 1. In the new Figure 2 we instead show the association of higher YKL-40 levels and shorter RTLs with the need for neonatal ventilatory support described as mechanical ventilation + CPAP more than one month (> 28 days).
It is our sincere hope that the reviewer will find these changes satisfactory. We believe that Figure 2 is now more coherent with the message in the abstract and again very grateful for the opportunity to improve our manuscript.